# Vertical versus horizontal Spatial-Numerical Associations (SNA): A processing advantage for the vertical dimension

**Luke Greenacre**[1], **Jair E. Garcia**[2], **Eugene Chan**[1,3], **Scarlett R. Howard**[4,5], **Adrian G. Dyer**[2,6¤]*

**1** Faculty of Business and Economics, Monash University, Caulfield East, Victoria, Australia, **2** Bio-Inspired Digital Sensing (BIDS) Laboratory, School of Media and Communication, RMIT University, Melbourne, Victoria, Australia, **3** Ted Rogers School of Management, Toronto Metropolitan University, Toronto, Canada, **4** Centre for Integrative Ecology, School of Life and Environmental Sciences, Deakin University, Burwood, Victoria, Australia, **5** School of Biological Sciences, Monash University, Clayton, Victoria, Australia, **6** Department of Physiology, Monash University, Clayton, Victoria, Australia

¤ Current address: Institut für Entwicklungsbiologie, und Neurobiologie, Johannes Gutenberg Universität, Mainz, Germany
* adrian.dyer@monash.edu.au

**Data Availability Statement:** All raw data as produced by DirectRT are available in the Dryad

## Abstract

Humans have associations between numbers and physical space on both horizontal and vertical dimensions, called Spatial-Numerical Associations (SNAs). Several studies have considered the hypothesis of there being a dominant orientation by examining on which dimension people are more accurate and efficient at responding during various directional SNA tasks. However, these studies have difficulty differentiating between a person's efficiency at accessing mental representations of numbers in space, and the efficiency at which they exercise motor control functions, particularly bilateral ones, when manifesting a response during an explicit directional SNA task. In this study we use a conflict test employing combined explicit magnitude and spatial directional processing in which pairs of numbers are placed along the diagonal axes and response accuracy/efficiency are considered across the horizontal and vertical dimensions simultaneously. Participants indicated which number in each pair was largest using a joystick that only required unilateral input. The experiment was run in English using Arabic numerals. Results showed that directional SNAs have a vertical rather than horizontal dominance. A moderating factor was also found during *post-hoc* analysis, where response efficiency, but not accuracy, is conditional on a person's *native* language being oriented the same as the language of the experiment, left to right. The dominance of the vertical orientation suggests adopting more vertical display formats for numbers may provide situational advantages, particularly for explicit magnitude comparisons, with some domains like flight controls and the stock market already using these in some cases.

Digital Repository at doi.org/10.5061/dryad.
0zpc866vj.

**Funding:** Funding: A.G.D. received funding support
from the Australian Research Council
(LE130100112), S.R.H. was supported by an Alfred
Deakin Research Fellowship. The funders had no
role in study design, data collection and analysis,
decision to publish, or preparation of the
manuscript.

**Competing interests:** I have read the journal's
policy and the authors of this manuscript have the
following competing interests: Adrian G Dyer
wishes to declare that he is an Academic Editor for
PLOS ONE.

## Introduction

Number sense requires the abilities to both establish the value/quantity represented by a number (often labeled numerical estimation) [1, 2] and to understand differences between numbers representing different values/quantities (often labeled magnitude comparison) [3]. There is evidence that this number sense has some spatial association, called Spatial Numerical Associations (SNAs), that arises for both non-symbolic and symbolic representations of numbers [4]. Research into directional SNAs has demonstrated that smaller numbers tend to be oriented towards left space and larger numbers towards right space along an apparent horizontal Mental Number Line [4–8]. Interestingly, this phenomenon does not seem to be limited to human subjects. Newly hatched chicks have also demonstrated a left to right bias when evaluating non-symbolic representations of number [9], while pigeons and blue jays appear to have either a left to right or right to left bias depending on the individual [10]. These results suggest that human SNAs may share a common basis with analogous associations in other vertebrates stretching back at least 310 million years [9, 10].

The directional nature of SNAs has been widely examined within the horizontal dimension, with substantive work also examining the vertical and to a lesser extent the frontal (near/far) dimensions [4, 8, 11–13]. Each dimension has been considered across several studies, with results often supporting that along the horizontal dimension larger numbers are rightwards [4, 11, 14], whilst along the vertical dimension larger numbers are upwards [15, 16]. When considering the frontal dimension, lower numbers are located closer to the subject with larger numbers increasing in magnitude with distance from the subject [8, 17]. There is evidence that all dimensions may manifest simultaneously to some extent with people mentally representing numbers along more than one dimension simultaneously [8], but the potential dominance of one dimension over others is a question just starting to be considered [17].

Research into directional SNAs involves evaluating how efficiently participants respond to number stimuli when those numbers are placed in different configurations in space. Greater efficiency in responses to certain configurations suggests that this spatial mapping is the more likely candidate for the inherent semantic representation of number in the mind. However, participants must exercise their motor control functions to manifest that response, be it clicking a button or looking at a number on a screen. Hence, a response to a number stimulus is a function of both a participant's efficiency at accessing mental representations of number in space, and the efficiency at which they exercise their physical motor control functions when manifesting a response to a stimulus [7, 18–22]. SNA effects found in the literature may be influenced by one or both of these factors [7, 19, 22]. Studies of the dominance of a particular dimension in number space, or even just across multiple domains of space, have thus far had particular difficulties disentangling these two factors, producing inconsistent results [8, 17].

One approach to measuring directional SNA effects across multiple domains is the multi-dimensional response box. In this device, buttons are placed along each of the tested dimensions (horizontal/vertical/frontal) with participants doing sequential tasks across each dimension (see for example Holmes [23]). The way the responses are elicited may produce a correlation among the dimensions though, as participants employ their left and right hands for all three dimensions repositioning them along dimensions in sequence. The right handedness of the typical subject may lead to common biases along all the dimensions based on the common inclination regarding where to place one's hands on each dimension, such as the right hand being placed far away on the frontal dimension as the dominant hand tends to be used to reach for objects (the definition of a dominant hand). Unique biases along particular dimensions may also arise where handedness is more compatible with responses along particular dimensions, likely the horizontal where the left and right hands naturally sit relative to the

body. These are problems common to bilateral button box type tasks, and the problems persist with even contemporary implementations of such devices [8]. A different approach is to use touch screen responses rather than button boxes. In one example, participants undertook a parity task (odd vs. even) along the horizontal, vertical, and diagonal (a conflict between horizontal and vertical) dimensions, with the horizontal dominating over the vertical [23]. Much like button boxes, though, participants tend to use both hands when responding. More contemporary methods have integrated virtual reality, allowing for more complex interactions with dimensions in space during experiments, but again elicit responses bilaterally through hand squeezes [24]. Bilateral responses likely confound potential spatial-numerical associations with handedness in response elicitation.

Alternative methods capture a participant's saccadic response during a task designed to elicit directional SNAs. There is a rich variety of visual spatial tasks that can be used examine directional SNAs, either with explicit or implicit number comparisons [25], and while many have been used to examine the horizontal dimension, far less has been done with the vertical [25], and nearly nothing in the determining of the dominance of one dimension. We found only one study that could be used to consider dominance; Hesse and Bremmer [16] captured saccadic response during a parity type task, an interesting approach that may circumvent the issues arising from handedness [26]. While not the focus of the analysis, there is some evidence suggesting that the horizontal dimension provides greater explanatory power than the vertical [16]. Vertical saccades do have longer latencies and lower peak velocities though [27], suggesting that vertical saccades require more effort than the horizontal saccades, again confounding results due to biases arising from response elicitation.

SNA tasks that capture head movement instead of eye movement may experience similar biases as head movement is often recruited to facilitate eye/gaze movement [28], suggestive of correlated bias for these two types of movement. However, the diverse range of muscle groups used in head/neck movement, particularly the likely use of different muscle groups to achieve pitch versus horizontal rotation [29, 30], and the individual differences in performance of various movement tasks makes bias difficult to predict [31, 32].

Our research thus seeks to account for motor control effects, in particular issues arising from bilateral response elicitation, when examining the hypothesis that there is a dominant dimension for directional spatial-numerical associations. In this study, we asked participants to select the larger number when presented with pairs of Arabic numerals on a screen using a joystick. We predict that if there is a dominant direction for SNAs, there would be significant differences in either accuracy or response time (efficiency) depending on the position of the largest number on the test screen. Alternatively, if there is no dominant dimension for directional SNA, we would observe no difference in either accuracy or reaction time wherever the largest number is presented vertically or horizontally on the screen.

## Method

To assess the hypothesis of humans having a dominant dimension for the accurate and efficient comparison of numbers (SNAs) we conducted a conflict test across the diagonal axis of a computer display. Such conflict tests pit the horizontal and vertical axes against each other, assuming that one can dominate over the other and that the diagonal itself is not an innate axis–all of which are consistent with prior research in this area [17]. Participants viewed pairs of numbers (Arabic numerals) along the multiple diagonal axes on a computer screen and indicated the location on the screen of the larger number, the number of greater magnitude, in each trial as quickly as possible using a joystick (Fig 1). Participants thus undertook explicit magnitude processing and explicit spatial directional processing when responding [14].

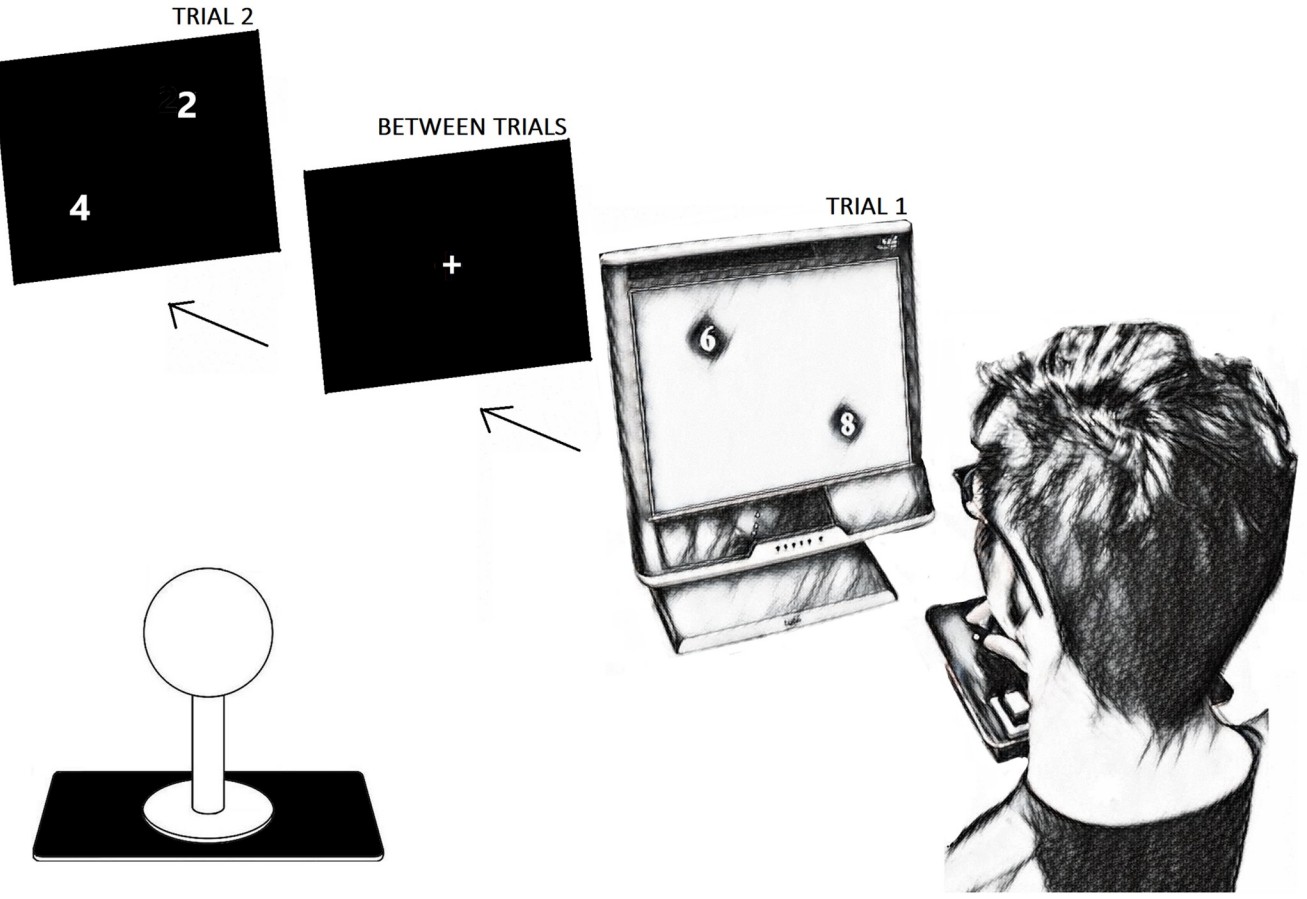

**Fig 1. Experimental setup as experienced by a participant.** As per the sketch on the right of the figure showing a view over a participant's shoulder, participants used the joystick placed in front of them to indicate the corner of the screen the larger of the two numbers (the target) appeared in each trial. A representative sequence of theoretical screens is also shown across the top of the figure, with the number pair 6,8 shown on the screen, followed by a centrally located (+) for 500 ms, then another number pair, which in this example is 4,2. The repeating pattern of a number pair (trials to which the participant responded with the joystick), then central (+), before further number pair trials comprised the main experimental task. Inset on the lower left of the figure is an indicative drawing of a ball-top joystick, as used by participants to give their response. The joystick allowed for any movement direction along the device's axial plane in two dimensions. Participants could hold the joystick in whatever grip style they felt was comfortable.

## Experimental design

The design arranged the numerals 1 through 9 into pairs using a full combinatorial. The pairs were repeatedly presented along the diagonals such that the larger (target) number in each pair appeared once in each of the four corners of the screen (Fig 1). The 36 number pairs arising from the combinatorial, and 4 possible corner locations for the target in each of those pairs resulted in 144 (= 36 x 4) experimental trials. The 'corner of the screen' for the placement of each numeral was defined as being 25% of the distance from the horizontal and vertical sides of the monitor. Trials were presented in pseudo-random order. Between trials a centrally located cross (+) was shown for 500 ms.

Participants indicated the location of the largest number on the screen for each trial. It is somewhat expected that the largest effects will be found when the larger number in the pair is above four, as this is beyond the subitizing range, in which processing is known to be highly accurate and rapid (2). With the chosen combinatorial the target will be five or more (hence, above the subitizing range) for 30 pairs, of which 10 will have both numbers in the pair being five or above. This provides substantial opportunity to detect differences in response accuracy

and reaction time. Having participants indicate the lesser of the two numbers in the pair would have yielded only 10 pairs where the target is five or more, providing little opportunity to detect differences in accuracy and reaction time, hence the response of the 'lesser' in each pair was not sought from participants.

## Experimental setup

Participants indicated their responses using a custom, zero-lag, ball-top joystick controlled by hand movements. This method has the advantage that cortical representations for the hand have been shown to be finely tuned to fundamental information processing across the body [33–35] and can thus permit good experimental access to test how participants optimally perceive stimuli. The joystick was programmed to have four detection points on each exact diagonal position (45˚ up-right, 135˚ down-right, 225˚ down-left, 315˚ up-left), and recorded a response when a diagonal position was triggered with the joystick. A hit involved the participant correctly indicating the position of the larger number (the target), with the angle of the joystick response then translated into perceived dominant preference for horizontal (left / right) or vertical (up /down) dimension. This translation involved coding the corner position to be a function of both the horizontal and vertical movement involved as separate variables. As per convention with ball top joysticks, and the playing of arcade games that use such devices, pushing the joystick forward indicated upwards and pulling the joystick indicated downwards. It was decided not to re-orient the joystick onto its side, with left, right, up and down then mapping directly (without translation) onto joystick movement. This decision was made as there would be a need to hold/hover the hand in space clasping the joystick to prevent inadvertent downward pressure on the joystick due to gravity. The need to hold/hover the hand would induce downwards bias in responses and could induce considerable fatigue in participants. Pilot experiments showed participants instantly adopted the spatial mapping framework implemented.

The joystick was placed wherever it felt comfortable for the participant. Participants were not provided specific instructions on how to hold the joystick, with any grip type comfortable for a participant considered to be acceptable, but participants were prompted not to rest their wrist on the unit or desk. This instruction was provided to reduce potential bias arising from restricting the potential movement of the hand on the joystick by having it rest on the unit and 'anchor' movements, while still taking advantage of its unilateral (single handed) response format. To facilitate compliance, fake buttons were installed on the joystick unit where the wrist would naturally rest and participants were told that pressing those buttons would invalidate the experiment. Participants could use either their left or right hand to hold the joystick. Responses were recorded using the DirectRT software (v2016, Empirisoft, USA).

Participants were seated at a desk in a curtained cubicle with their face 57 cm from the 17" computer display. A high-performance Tobii T120 monitor (Tobii, Sweden) was used as it had a 120 Hz refresh rate. The display had 1280x1024 pixel resolution and 338x270 mm visible display area (width x height). The numbers shown on the screen were 27 mm in height in Times New Roman font representing a visual angle of about 2.5˚, well above acuity limitations for the participant pool, and were shown in white with a black background.

## Sample

Participants (n = 73) were second year undergraduate students at a major Australian university. As the experiment was presented in English, the only recruitment criteria was that participants were fluent in English, regardless of whether they also spoke other languages. English fluency was a requirement of enrolment at the university. Participants were also required to

have a minimum 6:12 vision. All participants were confirmed to comply with this requirement either unaided or with correction from glasses/contact lenses via testing with a Snellen Eye Chart.

### Procedure

The sequence of events during the experiment were as follows. Participants arrived at the laboratory foyer and were provided with the explanatory statement to read. Prior to entering the lab their enrolment at the university was established (confirming English fluency) and their visual acuity was then assessed with a Snellen Chart. Upon entering the lab they were seated in a cubicle with the experiment set up on the computer ready to commence. Their distance from the monitor was measured and corrected as needed. The joystick was already in front of participants on arrival and they were instructed to move it to wherever comfortable for use. The experiment was then started in DirectRT. The participants followed the instructions on screen and undertook the experimental task. At the end of the task participants answered some demographic questions using a keyboard available. Participants then left the lab.

### Ethics

As noted above, participants were supplied with a printed explanatory statement to read. They indicated their consent to participate by pressing a button on the computer prior to continuing to the experimental task. This project and protocol were approved by the Monash University Human Research Ethics Committee. It was confirmed as conforming to the Australian National Statement on Ethical Conduct in Human Research. Project Reference Number 12214.

### Statistics

Data from each participant were recorded as individual.csv files, which were subsequently merged into a single file and imported into the R language and environment for statistical computing v 3.5.1 for analysis. All models reported in the paper were fitted using the glmer routine available in the package lme4. Models were tested using a likelihood ratio test using the anova command included in the base package of the software distribution.

## Results and discussion

### Demographics

Participants reported the demographic information at the end of the experiment, with 50.7% indicating they were male and 49.3% female, with their ages ranging between 18 and 23 years. Participants were asked about their experience with console video games, which often involve joystick operation, with about half of the sample (48.5%) reporting less than three hours of play per week and 10.6% reporting more than 18 hours per week. Regarding the handedness of participants, 87.7% reported that they write with their right hand, 94.5% reported generally throwing a ball with their right hand, and 90.4% reported doing this experimental task with their right hand. The alignment in these values indicates participants likely always opted to use their dominant hand in the experiment. Participants' *native* language was also captured as language orientation can vary from the Left-to-Right (LTR) orientation typical of English, the language employed in the experiment. The languages most frequently reported were English, Mandarin, Vietnamese, Cantonese, Indonesian, and Sinhalese; other reported languages included Luxembourgish, Dutch, Korean, Khmer, and Greek. Coding the languages according to World Wide Web Consortium (W3C) standards, which defines common standards for how

information is presented on the internet, it was found that 37% of native languages reported employ a mixture of script orientations, with the most commonly reported languages in this group being Mandarin, Cantonese and Korean, which may be presented in Left-to-Right (LTR) or Top-to-Bottom orientations when written. The majority of the sample (63%) reported a native language exclusively expressed in an LTR orientation, the most common being English followed by Vietnamese. Due to the proportion of participants having a native language that is not exclusively oriented LTR it was assessed as a potential moderator in *post-hoc* analysis.

## Proportion of correct choices

To test for an effect of dominant dimension in SNA on accuracy we formulated an initial linear model including as fixed predictors of the observed proportion of correct choices: the difference in magnitude between the two numbers (numerical distance), magnitude of the largest number, two categorical predictors with two levels each indicating the vertical (up/down) and horizontal (left/right) position of the largest (target) number, and an interaction term between the difference in magnitude and numerical distance. The initial model indicated that there was no significant interaction ($\chi^2 = 0.178$, df = 1, P = 0.673), so a reduced model was formulated excluding the non-significant interaction term.

The reduced model showed no significant difference when the largest number was viewed on the left or right ($\chi^2 = 2.20$, df = 1, P = 0.138), but the proportion of correct choices was significantly affected by the vertical position of the numbers ($\chi^2 = 56.6$, df = 1, P < 0.001). Specifically, the proportion of correct choices was greater when the larger number (the target) was positioned upwards (Fig 2, panels A-B). This shows evidence of a vertical dominance in the spatial numerical association, supporting the research hypothesis.

Interestingly, participant choices were also affected by the magnitude of the largest number ($\chi^2 = 78.6$, df = 1, P < 0.001) and the difference in magnitude with the smaller number (the distractor) ($\chi^2 = 129$, df = 1, P < 0.001). Participants' performance significantly improved when the difference in magnitude was larger and the target was above the 1–4 range. These results replicate the Numerical Distance Effect and Magnitude effects in number representations. The NDE has consistently shown that the larger the difference/distance between two numbers the better the performance when comparing them [3]. A possible reason for observing the effect around the 5–6 boundary line (Fig 2, panels A-B) is the lower 1–4 range is often associated with early subitizing ability when assessing quantities of objects, and those experiences of subitizing and the subsequent numeration of these quantities from an early age may provide a unique processing advantage.

## Reaction time

The same specification for the main model of correct responses was used for reaction time. The model indicates that response time was not significantly affected by either the horizontal ($\chi^2 = 1.13$, df = 1, P = 0.288) or vertical position of numbers ($\chi^2 = 1.42$, df = 1, P = 0.233). This result suggests that there is no evidence of a dominant dimension in spatial numerical associations when considering reaction time, providing no support for the research hypothesis for this response variable. We did observe, however, an interaction effect arising between magnitude difference and target magnitude ($\chi^2 = 26.1$, df = 1, P < 0.001). To understand the nature of this interaction we plotted the reaction time, as predicted by the model, as a function the largest number displayed on the screen for the differences in magnitude tested (Fig 3, Panel A). In all instances, response time decreased (i.e. Reaction speed increased) with the

## UP

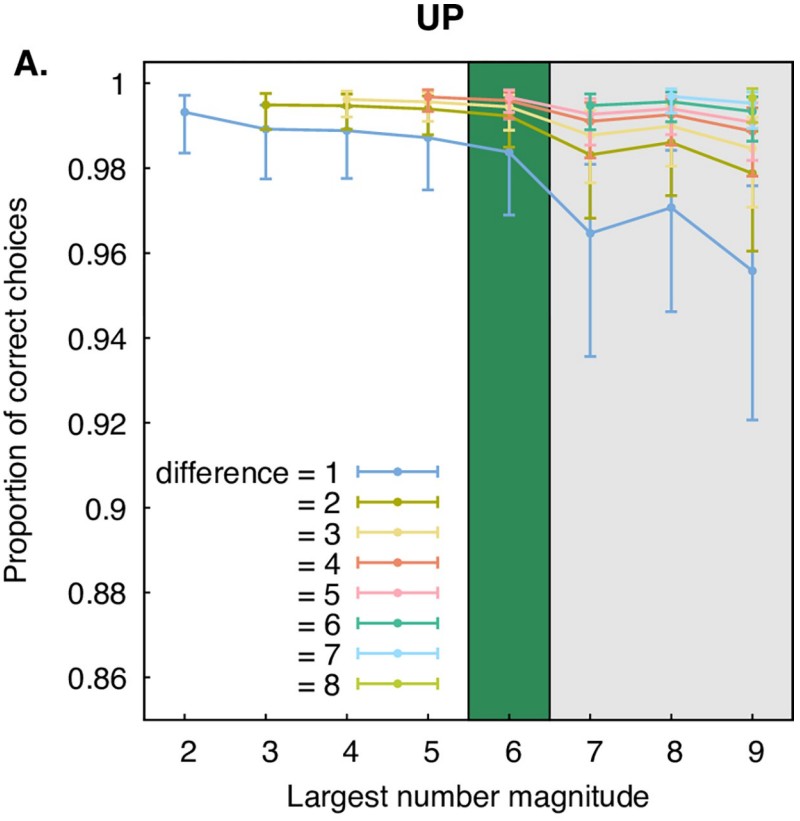

## DOWN

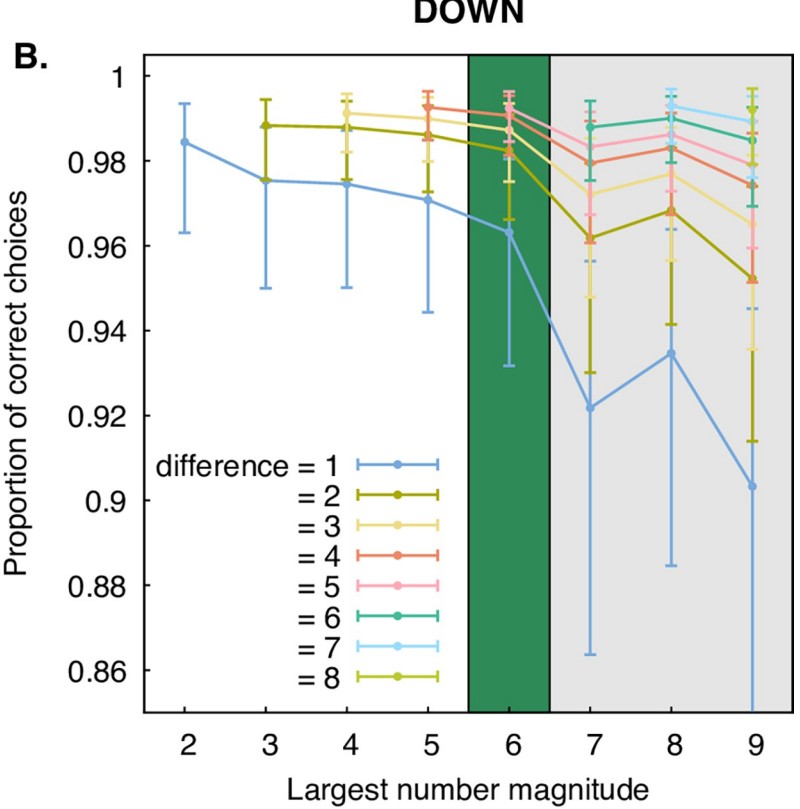

**Fig 2. Choice results.** The proportion of correct choices were higher when the largest number was upwards (panel A) versus downwards (panel B), supporting the research hypothesis. Additionally, the proportion of correct choices (panels A/B) for the largest number in a pair (1–9) increased with increasing numerical distance (colored lines). The shaded grey area indicates when both the target (larger) and distractor (smaller) numbers are above the 1–4 range, the shaded white area indicates when both are within the 1–4 range, and the green is the transition between these two areas. Error bars represent 95% CIs.

magnitude of the target and with the difference to the distractor, broadly in line with Numerical Distance and Magnitude Effects [3].

Since the seminal work on Spatial-Numerical Associations by Dehaene et al. 1993 [11], there has been a debate on the potential effect of directional reading habits on the left-right associations typical of the SNA effects, particularly, when considering bilingual participants [36]. Notably, in this study by Shaki and Fischer 2008 [36], although no differences in response time for SNAs were found in participants speaking Russian and Hebrew, the magnitude of this effect varied with language. Given that our sample included participants whose native language orientation is not exclusively expressed in a Left-to-Right (LRT) direction, we split the sample into two subsets, one whose native language is exclusively expressed in an LTR orientation and the other whose native language is not, and replicated the prior model for response time on each group separately as a *post-hoc* analysis.

For participants whose native language was oriented LTR (n = 47), results indicate a significant interaction between magnitude difference and target magnitude ($\chi^2$ = 21.9, df = 1, P < 0.001) on the participant's response time. This is the same as the main analysis of all participants. Different from the main analysis, however, we found a significant effect of vertical orientation on response time ($\chi^2$ = 4.60, df = 1, P = 0.032) with response time decreasing by 8.42 ms relative to the mean reaction time under the null hypothesis of 652 ms. That is, participants reacted faster when the larger (target) number was located on the upper section of the screen. As with the main model including all participants we found no significant effect of horizontal number location on response time ($\chi^2$ = 1.23, df = 1, P = 0.268). This lends further supports to the hypothesis of a dominant dimension for SNAs.

For participants whose native language was not exclusively LTR (n = 26), we found no significant effect of vertical orientation on response time ($\chi^2$ = 0.801, df = 1, P = 0.371), nor a significant effect of the horizontal number location ($\chi^2$ = 0.080, df = 1, P = 0.777). We found, however, a significant interaction between target magnitude and magnitude difference ($\chi^2$ = 5.19, df = 1, P = 0.023) as in the LTR subgroup.

Splitting the sample on this variable sheds light on the potentially important role of native language orientation on SNA effects in bilingual participants. We can see that the dominance of vertical processing is upheld in the response time data, for the LTR native language sub-set of the sample, whose native language orientation matched the orientation of the language used in the experiment (English). However, a different experimental design akin to the one used by Shaki and Fischer 2008 [36] and Fischer et al. 2009 [37] should be used to test that hypothesis.

Returning to the main model with the total sample, when examining variation in reaction time across all the possible differences in number magnitude between the target and distractor produced for each tested target magnitude (Fig 3, panels B-H), we obtained a more detailed view of the effects of these two variables on reaction time. When the largest number of a pair (the target) was smaller than or equal to five (Fig 3, panels B-D), median reaction time for the various differences in magnitude tested were close to the median reaction time for all observations (569 ± 78 median absolute deviation). As the magnitude of the largest number tested (the target) increased (Fig 3, panels E-H), we observed reaction times longer than the grand median. While not the explicit aim of this research, this result demonstrates that the

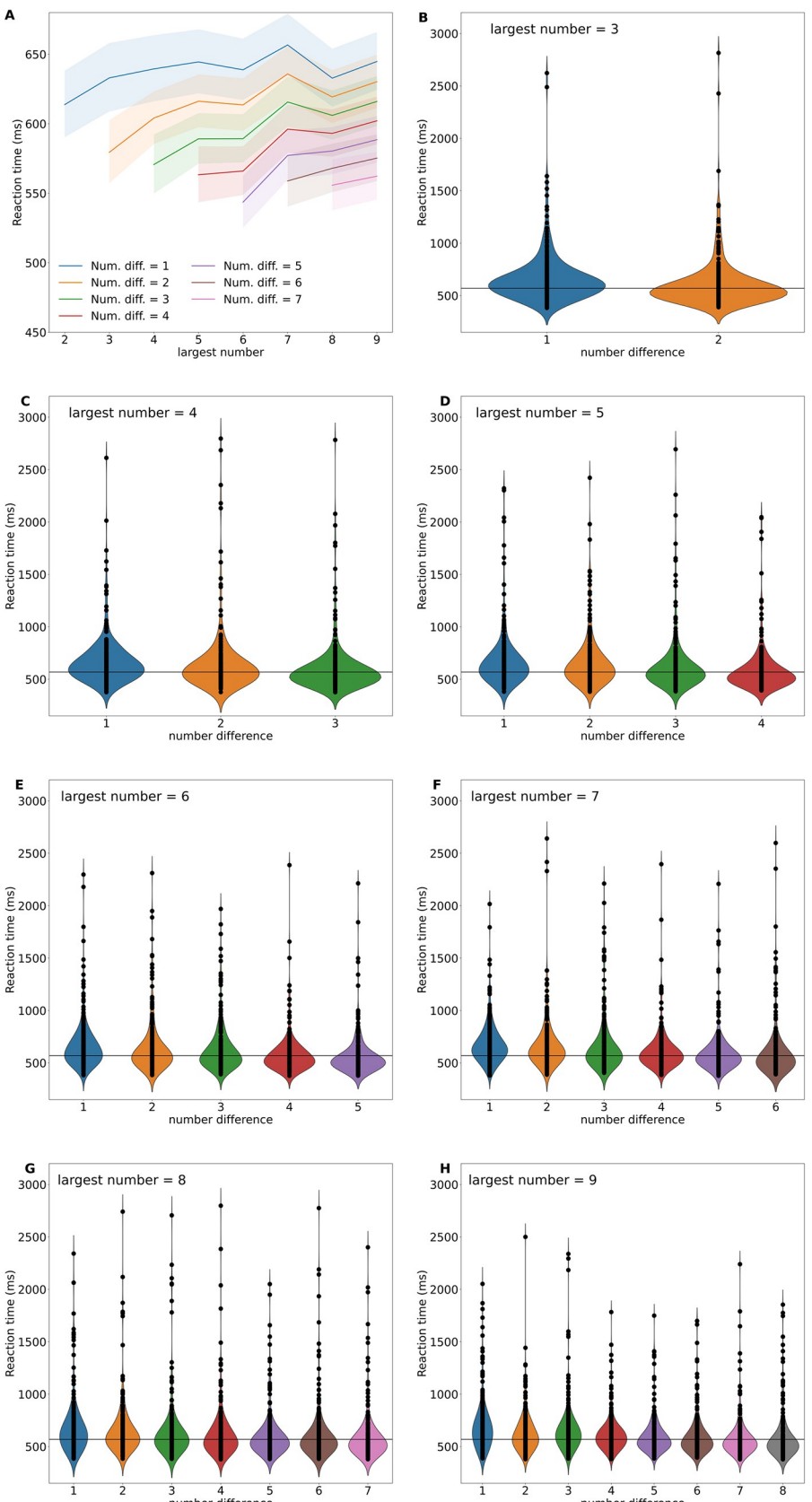

**Fig 3. Effect of target number magnitude, and difference in magnitude between target and distractor numbers (numerical distance) on reaction time.** Panel A: The graphical representation of the statistical model used for analyzing the participants' reaction time. Reaction time increased (i.e. participants took more time to respond) when the difference in magnitude between the numbers decreased, being slowest to react for a difference of 1 unit, and fastest to react for a difference of 7 units as indicated by the respective coloured lines. For any given difference in magnitude, reaction time also increased (i.e. participants responded more slowly) with the increasing magnitude of the largest number (the target) in the pair (values shown on the x-axis). Panels B-D show the distribution of the observed reaction times (black circle markers) for the various differences in number magnitudes tested for each target number magnitude as 'violins' whose peak represents the median value for each target magnitude and difference combination. The horizontal line in panels B-H represent the median of all observed reaction times. In all instances, reaction time increases (i.e. participants responded more slowly) with the increasing magnitude of the largest number of the pair, but diminishes with increasing difference in magnitude between the two numbers (target and distractor) in the pair displayed.

differences in magnitude (numerical distance) and number magnitude effects have a relationship, a relationship that has only recently been explored [38]. When the reaction time for the various differences in magnitude tested for each target magnitude were considered individually, we always observed a reduction in reaction time with increasing differences consistent with the Numerical Distance Effect.

## Conclusions

This research hypothesized that a particular dimension was dominant (either horizontal or vertical) for directional spatial-numerical associations (SNAs). To establish if such a dominance may exist, the method used an explicit magnitude comparison task with an explicit directional component. The method removed the biases arising from bilateral and similar response formats by employing a unilateral joystick apparatus. The results generally supported that numbers are responded to more accurately and more quickly (efficiently) when a larger number is in the upwards vertical position with the horizontal position having no impact on response accuracy nor efficiency, suggesting that the dominant mental representation of number is vertical not horizontal.

This is one of, if not the, first experiment that has used explicit number comparison and explicit directionality to establish the dominance of the vertical dimension. The results here are supported by research in the implicit paradigm. Most notably the dominance of the vertical dimension was recently found in a similar conflict task that employed implicit directionality, and in line with our results, found larger numbers were more efficiently responded to when upwards [39]. Notably, they too found no effect for the horizontal dimension at all when the vertical was simultaneously assessed [39].

Interestingly, some environments like the stock market and aircraft flight controls already utilize vertical displays of numbers, and there may be benefits to other fields of human endeavor adopting such display formats in which situational advantages may arise [22]. Our research thus suggests a need for more research on the advantages of *vertical* number processing, and how or why vertical processing may have evolved from our phylogenetic roots to dominate over horizontal number processing.

A limitation of this work arises from the challenge of testing vertical orientations in response tasks. We needed to translate the 'vertical' response of participants from their movement of the joystick forwards (up) and backwards (down). We could not orient the joystick onto its side to create an actual up and down movement, as participant responses would have been influenced by the likely fatigue and bias induced from having the resist gravity when using the joystick. The need to translate forward and backward to representing up and down, and not having to translate for left and right movements presents a practical limitation of the apparatus. While the apparatus overcomes bilateral response bias, future researchers need to

accommodate this limitation and work around it when necessary. This limitation could prove problematic in some paradigms, particularly those where the near/far and vertical dimensions are both relevant.

An interesting potential boundary condition to the dominance finding was identified in the analysis. While the vertical dominance was readily apparent in the proportion of correct choices (accuracy), it only manifested in the reaction times (efficiency) of those whose native language orientation matched the LTR orientation of English, the language used in the experimental task. While all participants were fluent English speakers, native language orientation appears to play some role in moderating SNA effects. Such moderating effects have been noted among bilinguals [36], leading to questions regarding whether the habits formed during the learning of one's native language may diminish SNA effects when tested in languages with different script orientations [37]. Further research is needed on the role of native language and/or the habituations typical of such learned behaviors on SNAs. Indeed, other habituations, such as counting direction [37], may also play a role.

The data also showed evidence of numerical distance and magnitude effects. Both effects have been found in numerous contexts, and generally describe the greater efficiency at which humans compare numbers that are further apart (of greater distance), and those that are smaller (especially those within the subitizing range) [38]. Finding such effects in our data is not surprising, as they manifest in a broad range of conditions, but do assist in validating the method and joystick apparatus as being suitable for research into numerical cognition. Validating the usefulness of the joystick apparatus is particularly important is it helps us disentangle the efficiency at which people access mental representations of number in space, and the efficiency at which they exercise physical motor control functions when responding to stimuli [7, 21, 22] by employing a unilateral rather than bilateral response mode.

## Acknowledgments

We thank Harmen Oppewal and the Monash Business Behavioral Laboratory for facilities and support.

## Author Contributions

**Conceptualization:** Luke Greenacre, Jair E. Garcia, Eugene Chan, Adrian G. Dyer.

**Data curation:** Luke Greenacre, Jair E. Garcia, Eugene Chan.

**Formal analysis:** Luke Greenacre, Jair E. Garcia, Eugene Chan.

**Funding acquisition:** Luke Greenacre, Adrian G. Dyer.

**Investigation:** Luke Greenacre, Eugene Chan.

**Methodology:** Luke Greenacre.

**Project administration:** Luke Greenacre.

**Resources:** Luke Greenacre.

**Software:** Luke Greenacre.

**Validation:** Luke Greenacre, Jair E. Garcia, Eugene Chan, Adrian G. Dyer.

**Visualization:** Jair E. Garcia, Eugene Chan.

**Writing – original draft:** Luke Greenacre, Jair E. Garcia, Eugene Chan, Scarlett R. Howard, Adrian G. Dyer.

**Writing – review & editing:** Luke Greenacre, Jair E. Garcia, Eugene Chan, Scarlett R. Howard, Adrian G. Dyer.

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
