## [Decision Letter · Decision Letter 0]

28 Mar 2022

PONE-D-21-40286The Vertical versus Horizontal SNARC Effect: A processing advantage for being upwardsPLOS ONE

Dear Dr. Dyer,

Thank you for submitting your manuscript to PLOS ONE. After careful consideration, we feel that it has merit but does not fully meet PLOS ONE’s publication criteria as it currently stands. Therefore, we invite you to submit a revised version of the manuscript that addresses the points raised during the review process.

I would first like to apologize for the delay in making a decision regarding your manuscript. I was waiting for the review of an additional expert who finally could not complete the assignment. I am therefore making a decision based on the two reviews I was able to secure and my own reading of the manuscript. As you can see in the two reviews at the bottom of this email, the reviewers disagree on whether the manuscript is suitable for publication in PLOS ONE. While the first reviewer suggests a rejection, the second reviewer is more positive and points out that your manuscript might make an interesting contribution to the literature. Based on these reviews and my own reading of the manuscript, I am willing to give you the opportunity to revise and resubmit the paper. However, I urge you to carefully consider the different points raised by the reviewers. Most notably, both experts point out that a shortcoming of your task is that participants were only asked to select the largest (and not the smallest) number. You therefore either need to convincingly explain why this task feature did not affect your results, or (preferably) run an additional experiment in which you present participants with the missing condition. Another important issue raised by reviewer #1 concerns the heterogeneity in native languages of the participants. You also need to carefully address this point, possibly with additional analyses. Finally, both reviewers suggest changes in the structure of the manuscript, which I strongly encourage you to take into consideration.  

We look forward to receiving your revised manuscript.

Kind regards,

Jérôme Prado

Academic Editor

PLOS ONE

“We thank Harmen Oppewal and the Monash Business Behavioral Laboratory for facilities and support; Funding: A.G.D. received funding support from the Australian Research Council (LE130100112), S.R.H. was supported by an Alfred Deakin Research Fellowship”

“A.G.D. received funding support from the Australian Research Council (LE130100112), S.R.H. was supported by an Alfred Deakin Research Fellowship.”

“A.G.D. received funding support from the Australian Research Council (LE130100112), S.R.H. was supported by an Alfred Deakin Research Fellowship.”

Reviewers' comments:

Reviewer's Responses to Questions

**Comments to the Author**

1. Is the manuscript technically sound, and do the data support the conclusions?

Reviewer #1: No

Reviewer #2: Partly

2. Has the statistical analysis been performed appropriately and rigorously? 

Reviewer #1: No

Reviewer #2: Yes

3. Have the authors made all data underlying the findings in their manuscript fully available?

Reviewer #1: Yes

Reviewer #2: Yes

4. Is the manuscript presented in an intelligible fashion and written in standard English?

Reviewer #1: Yes

Reviewer #2: Yes

5. Review Comments to the Author

Reviewer #1: General comment

The aim of the present study was to investigate whether there is a dominant dimension (horizontal vs vertical) in Spatial Numerical Association of Response Codes (SNARC). The task was a comparison task where pairs of numbers are placed along the diagonal axes and the position of the largest number (the target) must be identified. The results show that participants respond more efficiently in vertical than horizonal orientations when both numbers being compared are beyond the range of 1-4; furthermore, when the target is in the upward position the response time decreasing significatively, but only in a subset of data including responses from participants whose language was not exclusively expressed in a left to right orientation (E.g. Cantonese and Korean).

I think that the manuscript has several methodological issues.

The type of task used is interesting and could add to the literature concerning this line of research. However, the sample is unbalanced as it includes 63% of participants whose native language exclusively uses a left-to-right (LTR) oriented script and 37% of participants whose native languages employed a mixture of script orientations which may be presented LTR or Top to Bottom. The authors should have used a sample composed only of participants who exclusively use a LTR; alternatively, if the authors wanted to compare different reading-writing directions groups, the sample had to be balanced 50% / 50%. Their results and conclusions are affected by their sampling.

Secondarily, in its present form, the study seems "incomplete” as participants were always asked to identify only the largest number of the pair (the target); to be in line with SNARC effect, the authors should also have included in the study responses when the target was the smallest number.

In general, the article not well-structured (for example the hypotheses of the study are not defined) and contains, in addition to the aforementioned methodological problems, many parts to be rewritten and better explained.

Please find below some comments concerning other critical aspects (major points only).

Detailed comments

Abstract

- P3, line 49-51 : What does the task consist of? the description is not clear; please specify.

- P3, line 51-53: “Results show that humans are faster at number comparisons in vertical rather than horizontal orientations, but only when both numbers being compared are beyond the range of 1-4 and the largest number is in the upward position”. The results indeed shows that the participants when both numbers being compared are beyond the range of 1-4 respond more efficiently (the proportion of correct choices is higher) but not faster. Furthermore, the faster response when the target was in the upward position was found only with a subset of participants. Please correct the inaccuracy.

- P4, line 65-69: The SNARC effect is not alone the association between numbers and space along a mental number-line oriented from left to right with small number on the left (e.g. 1,2) and large numbers on the right (eg. 8,9); in fact, it refers also to the phenomenon that individuals typically react faster to relatively smaller numbers with left-sided responses and faster to relatively larger numbers with right-sided responses. The authors should add this description.

Introduction

- P6, line 135: Which is the “innate biases across dimension”?

Method

- P8, line 162-164: How the angle of the joystick response was translated into perceived dominant preference for horizontal (left / right) or vertical (up /down) orientation?

- P8, line 185: I suggest replacing the word “people” with participants.

Results and Discussion

- P9, line 201: The authors should divide the results and discussion; furthermore, the manuscript does not present a proper discussion.

- P11, line 239-242: the meaning of this section in unclear; it should be rewritten more clearly.

- P12, line 268: “special-numerical associations” makes little sense; please, replace with “spatial-numerical associations”.

- P12, line 274-275: “Results show that numbers are responded to more efficiently when a larger number is in the upwards vertical position with the horizontal position having no impact on response efficiency, suggesting that the dominant mental representation of number is vertical not horizontal”. It is wrong to say that in the numerical representation the dominant dimension is the vertical as the results were found only in a subset of participants (people whose native languages employed Left to Right or Top To Bottom script orientations) furthermore, the significant effect in this subset of participants could be due to the native language of this participants.

Reviewer #2: Dear Authors,

I am pleased to review a manuscript that I consider informative in the field of spatial-numerical associations (SNAs). I recommend this article for publication as it brings a valid methodological and theoretical contribution but only after making major modifications and clarifications raised by the comments below. I list them following the order of the article’s main sections.

ABSTRACT

In the Abstract, the number comparison task is not defined (e.g., the task required participants to indicate which of two, visually displayed, single-digit numbers was larger).

INTRODUCTION:

1. In order to better contextualize the authors’ contribution in the field of SNAs, I would suggest them to use the recent terminology coined by Cipora and colleagues (2020; doi: 10.3389/fpsyg.2020.00182) and by Shaki and Fischer (2018; doi.org/10.1016/j.cognition.2018.02.022). Indeed, with the authors’ novel method described in this study, they aim to assess “Direction SNAs” during “explicit magnitude processing” and “explicit spatial directional processing”;

2. In order to give a more comprehensive overview of the literature, I recommend the authors to consider the following two articles:

- Sixtus, E., Lonnemann, J., Fischer, M. H., & Werner, K. (2019). Mental number representations in 2D space. Frontiers in psychology, 10, 172. https://doi.org/10.3389/fpsyg.2019.00172. This study assessed “Direction SNAs” during “implicit magnitude processing” and “implicit spatial directional processing” along the horizontal and vertical axes;

- Lohmann, J., Schroeder, P. A., Nuerk, H. C., Plewnia, C., & Butz, M. V. (2018). How deep is your SNARC? Interactions between numerical magnitude, response hands, and reachability in peripersonal space. Frontiers in psychology, 9, 622. https://doi.org/10.3389/fpsyg.2018.00622. This study assessed interactions between horizontal and radial spatial-numerical mappings in a virtual reality environment;

- Felisatti, A., Ranzini, M., Blini, E., Lisi, M., & Zorzi, M. (2022). Effects of attentional shifts along the vertical axis on number processing: An eye-tracking study with optokinetic stimulation. Cognition, 221, 104991. https://doi.org/10.1016/j.cognition.2021.104991. This study revealed bi-directional links between numbers and attentional orienting along the vertical axis.

3. I recommend the authors to explicitly describe the hypotheses.

METHOD:

1. Even if spaces are used to separate different subsections, I would suggest the authors to use also different subtitles. These would enable the reader to better organize the methodological and conceptual parts.

2. Participants: Did the authors record information about the handedness of the participants and their familiarity with joystick-related activities and games?

3. Cultural influence: Evidence has shown that not only reading direction habits but also counting direction habits play a role in spatial-numerical associations. I suggest the authors to consider this aspect (Shaki, S., Fischer, M. H., & Petrusic, W. M. (2009). Reading habits for both words and numbers contribute to the SNARC effect. Psychonomic bulletin & review, 16(2), 328-331. https://doi.org/10.3758/PBR.16.2.328);

4. Material: A picture of the setting would be useful to visualize the participant-joystick interaction;

5. Sequence of event: A picture showing the timeline and the sequence of event would be informative;

6. Task confound: An additional block (counterbalanced within or even between participants) asking participants to indicate the smaller and not the larger between the two numbers would have been very informative. Indeed, the unique association between large numbers with upward space might also be related to focus only on the relative larger numbers;

7. Spatial confound: Since the main aim of the study was to control for methodological confounds related to motor response, I would have expected a method without any spatial connotations. Instead, in the study both the stimulus encoding and the response were lateralized. Moreover, the device entailed confounds between the radial and the vertical dimensions. I suggest the authors to report more in detail the reasons motivating each methodological decision.

RESULTS AND DISCUSSION

1. I suggest the authors to subdivide the different sections by adding subtitles (e.g., Accuracy preprocessing and results, Reaction time preprocessing and results, Discussion);

2. The Discussion part is not comprehensive. I recommend the authors to further elaborate it by interpreting the results in light of the literature considered in the Introduction. Moreover, I would suggest them to add a “Limitation and future directions” section where to report the confounds/limitations of their method and their solutions to overcome them in future research.

6. PLOS authors have the option to publish the peer review history of their article (what does this mean?). If published, this will include your full peer review and any attached files.

Reviewer #1: No

Reviewer #2: No

---

## [Author Response · Author response to Decision Letter 0]

19 May 2022

We have uploaded a formatted and detailed Rebuttal letter with a Table explaining all revisions relative to reviewer advice; and we have provided a track changes version logging these changes.

---

## [Decision Letter · Decision Letter 1]

27 Jun 2022

PONE-D-21-40286R1Vertical versus Horizontal Spatial-Numerical Associations (SNA): A processing advantage for the vertical dimensionPLOS ONE

Dear Dr. Dyer,

Thank you for submitting your manuscript to PLOS ONE. I have sent it to a reviewer of the previous version. As you can see below, the reviewer thinks that the manuscript is much improved. I also think that you addressed the previous comments satisfactorily and am happy to accept the manuscript for publication in PLOS ONE, pending the minor modifications recommended by the reviewer.  Please submit your revised manuscript by Aug 11 2022 11:59PM. If you will need more time than this to complete your revisions, please reply to this message or contact the journal office at plosone@plos.org. Please include the following items when submitting your revised manuscript:A rebuttal letter that responds to each point raised by the academic editor and reviewer(s). You should upload this letter as a separate file labeled 'Response to Reviewers'.A marked-up copy of your manuscript that highlights changes made to the original version. You should upload this as a separate file labeled 'Revised Manuscript with Track Changes'.An unmarked version of your revised paper without tracked changes. You should upload this as a separate file labeled 'Manuscript'.If applicable, we recommend that you deposit your laboratory protocols in protocols.io to enhance the reproducibility of your results. Protocols.io assigns your protocol its own identifier (DOI) so that it can be cited independently in the future. For instructions see: https://journals.plos.org/plosone/s/submission-guidelines#loc-laboratory-protocols. Additionally, PLOS ONE offers an option for publishing peer-reviewed Lab Protocol articles, which describe protocols hosted on protocols.io. Read more information on sharing protocols at https://plos.org/protocols?utm_medium=editorial-email&utm_source=authorletters&utm_campaign=protocols.

We look forward to receiving your revised manuscript.

Kind regards,

Jérôme Prado

Academic Editor

PLOS ONE

Journal Requirements:

Reviewers' comments:

Reviewer's Responses to Questions

**Comments to the Author**

1. If the authors have adequately addressed your comments raised in a previous round of review and you feel that this manuscript is now acceptable for publication, you may indicate that here to bypass the “Comments to the Author” section, enter your conflict of interest statement in the “Confidential to Editor” section, and submit your "Accept" recommendation.

Reviewer #2: All comments have been addressed

2. Is the manuscript technically sound, and do the data support the conclusions?

Reviewer #2: Yes

3. Has the statistical analysis been performed appropriately and rigorously? 

Reviewer #2: Yes

4. Have the authors made all data underlying the findings in their manuscript fully available?

Reviewer #2: Yes

5. Is the manuscript presented in an intelligible fashion and written in standard English?

Reviewer #2: Yes

6. Review Comments to the Author

Reviewer #2: Dear Editor and Authors,

I am happy to endorse the publication of the article "Vertical versus Horizontal Spatial Numerical Associations (SNAs): A processing advantage for the vertical dimension". The Authors addressed all the points raised by me in a comprehensive way, adding conceptual and methodological clarifications. However, I would recommend the Authors to make few more modifications:

1) HYPOTHESIS: Specify that they predict prevalence of one dimension and motivate it, for instance by taking into account the Hierarchical view (Fischer, 2012), according to which the vertical dimension is considered the most grounded and universal one;

2) METHOD: Explicitly report in the text the reasons why they did not include the block involving response to the smaller number. I suspect that many readers would have the same question;

3) DISCUSSION: Insert a paragraph describing the limitations of the study.

7. PLOS authors have the option to publish the peer review history of their article (what does this mean?). If published, this will include your full peer review and any attached files.

Reviewer #2: No

---

## [Author Response · Author response to Decision Letter 1]

14 Jul 2022

We have provided a detailed response in Table for in the supplied file "PLOS Response to reviews_agd.docx". The table form does not translate to inline text well.

---

## [Editor Report · Decision Letter 2]

9 Aug 2022

Vertical versus Horizontal Spatial-Numerical Associations (SNA): A processing advantage for the vertical dimension

PONE-D-21-40286R2

Dear Dr. Dyer,

We’re pleased to inform you that your manuscript has been judged scientifically suitable for publication and will be formally accepted for publication once it meets all outstanding technical requirements.

Kind regards,

Jérôme Prado

Academic Editor

PLOS ONE
---

## [Editor Report · Acceptance letter]

16 Aug 2022

PONE-D-21-40286R2 

Vertical versus Horizontal Spatial-Numerical Associations (SNA): A processing advantage for the vertical dimension 

Dear Dr. Dyer:

I'm pleased to inform you that your manuscript has been deemed suitable for publication in PLOS ONE. Congratulations! Your manuscript is now with our production department. 

Kind regards, 

on behalf of

Dr. Jérôme Prado 

Academic Editor

PLOS ONE